# On the Application of Developmental Cognitive Neuroscience in Educational Environments

**DOI:** 10.3390/brainsci12111501

**Published:** 2022-11-04

**Authors:** Gerry Leisman

**Affiliations:** 1Movement and Cognition Laboratory, Department of Physical Therapy, University of Haifa, Haifa 3498838, Israel; g.leisman@alumni.manchester.ac.uk or g.leisman@edu.haifa.ac.il; 2Department of Neurology, Universidad de Ciencias Médicas de la Habana, Havana 11300, Cuba

**Keywords:** cognitive neuroscience, brain, learning, education, embodiment, movement, optimization

## Abstract

The paper overviews components of neurologic processing efficiencies to develop innovative methodologies and thinking to school-based applications and changes in educational leadership based on sound findings in the cognitive neurosciences applied to schools and learners. Systems science can allow us to better manage classroom-based learning and instruction on the basis of relatively easily evaluated efficiencies or inefficiencies and optimization instead of simply examining achievement. “Medicalizing” the learning process with concepts such as “learning disability” or employing grading methods such as pass-fail does little to aid in understanding the processes that learners employ to acquire, integrate, remember, and apply information learned. The paper endeavors to overview and provided reference to tools that can be employed that allow a better focus on nervous system-based strategic approaches to classroom learning.

## 1. Introduction

### Cognitive and Educational Neurosciences in Co-Development: Seeking a Common Language

Cognitive Neuroscience and Human Factors Engineering have made many strides over the past eighty years and the application to classroom-based instruction offers the possibility of a fundamental sea change in how the educational product is delivered and acquired. Recent advances in developmental cognitive neurosciences have produced fundamental changes in how it is that we understand nervous system structure and function as applied to thinking, cognition, memory, brain organization and behavior and much more, than previously thought. We now know that simplistic left-right differences and cerebral asymmetries are less important in understanding classroom learning but more multifaceted brain network, applications to instruction and external means of altering brain chemistry and neuroplasticity to facilitate learning have led to newly developed concepts and findings that have not, as yet, found their way into the classroom and in teacher training and in educational policy [1,2,3].

We require the advancement of innovative models to better understand activities that can importantly affect motivation, learning, and memory as well as evaluation methodologies that can observe, study and assess these functions. We are slowly realizing that there exists a significant intersection between the problems of psychological, sociological, and educational processes and those of neurobiology, biochemistry, and neurophysiology, with the possibility of reciprocal assistance [4,5]. Neuroscience has influenced school-based activity in various ways. For example, it has provided us with a better understanding of the nature of dyslexia [4] and has offered [5] and insights into how diverse variables such as attention, sleep, relationships and anxiety can affect educational outcomes [6,7]. Many difficulties exist in actualizing cognitive neuroscience applied to the classroom. Principally the various disciplines have different end goals such as prescriptive v. descriptive or fact v. solution oriented. Additionally, the neurosciences have been measuring. effects in milliseconds to minutes whereas education has been more concerned with changes measured in days, weeks, and years [8,9].

With this said, there does exist a knowledge base in the developmental cognitive neurosciences that can justify its translation into practical classroom solutions [1,2,3,6,7,8] and this we will provide in greater detail later in the paper. As examples, we know that: (a) teachers do not convey knowledge to learners as learning dynamically builds and rebuilds neural networks [6]; (b) a learner’s performance is highly dependent on the context in which the information is provided [10] and therefore, (c) learners require the ability to create the contents in which to facilitate their own learning [11]. Additionally, (d) learned skills and functions are not separate and isolated and learned in a linear fashion, but rather they consist of networks of interrelated functions in which complexity rather than difficulty should be emphasized [12]. Furthermore, learning both concepts and behaviors is predicated on the coordination and building of basic skills that will parlay into increasingly more complex learned behaviors [13] and, (e) regression is a necessary feature of learning [14].

The internet has also provided us with the ability, using the knowledge-based of cognitive neurosciences, to create new models of human cognitive development that ought to amend our understanding of the learning process and how training is provided. There exists a relationship between context and performance [15] and from reflexes to abstract thinking with basic skills that integrate within more complex skills [16]. As a result, we often see skills regression with significant implications for how school programming and curriculum are implemented and assessed. The current but old linear models, consisting of the training of isolated skills, treat and measure learning success as a steady rise. It is even the basic assumption of IQ testing [17]. We will see how we likely will fare better by focusing less on answers to questions and more on why and how learners learn in the way that they do. Grading systems reflect the current thinking in education that certainly does not account for the necessary regression of skill. If assessment is continuous, for example, rather than provided at some endpoint, regression could then be accounted for with skills assessment then being individualized, functioning then in a process-oriented rather than linear fashion.

This paper proposes that: (A) The understanding and evaluation of interregional communication in the brain in necessary. (B) While probably still necessary, the employment of standardized achievement and aptitude tests is not sufficient and (C) “cognitive efficiency” is likely a better evaluative model for employment is schools than the current evaluation of mastery v. non-mastery of learned material. While it may not be palatable for educators to consider early childhood education in such a way, they are producing a product and production management techniques should be a useful and effective means for evaluating not just the product but the “manufacturing process” of that product as well.

## 2. Neuroanatomical and Neurophysiological Relevancies and Irrelevancies in Learning and Classroom Settings

### 2.1. Brain Anatomy Is Irrelevant to Educational Practice but Not Functional Connectivity

Both adult and child learners possess, significant degrees of localized brain function which is inadequate to understand neuroplasticity, neural regeneration, spontaneous recovery, or optimized performance neurologically and cannot adequately explain its translation into school-based practice. Alternatively, achievement and aptitude testing are he currently employed tools for evaluating educational gains [18,19]. Achievement testing are concerned with the evaluation of educational gains designed to evaluate teaching and learning effectiveness based on current operational definitions that do not necessarily account for who brain functionality Such tests do not necessarily evaluate either the optimized function of the learner or for that matter the teacher and curriculum. Aptitude testing largely deals with the probability of achievement within the parameters of current educational practice but does not provide a comprehensive view of the tool skills, both physiological and cognitive necessary for optimized function of both the learner and the instructor.

These problems should not be a surprise. Developmental cognitive neuroscience is a newly developing field that as a consequence of its newness has generated numerous hypotheses that can appear fragmentary in nature. For us to apply sometimes disparate hypotheses into practice could be confusing at best and potentially damaging at worst. We, however, are endeavoring, not to throw the baby out with the bathwater. With advances in imaging sciences, genetic, and electrophysiological approaches currently available we have begun to better comprehend surprising new findings applied to new situations including the classroom [20]. It is suggested that the reader refer to the theoretical review and hypotheses drawn by Mark Johnson who has covered the principal arguments in this burgeoning field copiously [20].

In endeavoring to appreciate why neuroanatomic conceptualizations of the learner is not singularly relevant for education, it is essential to understand that what we are endeavoring to accomplish in educational practice. We are attempting to understand the neurological basis of cognitive development but not by which region of the brain controls a particular cognitive skill, but rather how efficiently it is functioning and networking [21]. While not the scope of this paper to provide a detailed overview of this principle, the reader is invited to review these concepts more comprehensively elsewhere [22].

To illustrate the importance of functional organization rather than localization of function, Figure 1A below-right, presents a CT-Scan of the brain of a patient T.S. while in a persistent vegetative state and 1A-left is a CT of a normal individual provided for comparison purposes. Figure 1B is of an otherwise normal individual of normal intelligence born with hydrocephalus, with clear functional differences noted between individual T.S. and the individual with congenital hydrocephalus.

The concept of “cortical efficiency” [23,24,25,26,27] suggests that greater skill in the performance of a cognitive task is associated with more optimized neural information processing and not necessarily the brain region concerned with that processing. Intuitively, one might presume that the performance of complex cognitive tasks would be more highly associated with greater brain activity. For the cerebral cortex, the contrary is so. Greater performance in numerous tasks, including [28,29,30,31] numeric, figural [30,31,32,33] and spatial reasoning [34] are associated with diminished energy consumption in numerous cortical regions. This has also been examined electroencephalographically in different frequency bands. The amount of a background power (7.5–12.5 Hz) decreases during cognitive activity in comparison to resting state. This reduction has been found to be related to significantly higher performance in subjects with superior IQ scores [35] or with higher performance after training on complex tasks, indicating a more efficient processing strategy for cognitive tasks [28].

### 2.2. Plasticity and Connectivity in Brain Networking: A Basis for Child and Adult Education

In order to better understand the importance of networks rather than hemispheric specialization, lateralization or neuroanatomy, it might be useful to propose a gedanken experiment. If one were to completely remove the posterior visual areas of a hemisphere in a cat (including parietal regions, that cat would be rendered profoundly and permanently unresponsive to visual stimuli in half of visual space opposite to the site of the cortical removal. Such an animal would be rendered blind as with radical damage to human geniculostriatal system. By inflicting additional damage on such a severely impaired animal at midbrain one would restore ability to orient and localize stimuli in formerly blind field. This would be accomplished by removing the contralateral superior colliculus or by severing fibers in the central portion of the collicular commissure. In other words, adding damage in the brainstem to the cortical damage “cures” the behavioral effect of massive cortical damage. The Sprague effect is a consequence of secondary effects generated at the brainstem level by unilateral cortical removal. The damage deprives the ipsilateral superior colliculus of its normal cortical input. Damage unbalances collicular function via indirect projection pathway, chiefly the substantia nigra to the colliculus, which crosses the midline in a narrow central portion of the collicular commissure. The “restorative” interventions in part adjust this imbalance permitting the collicular mechanism to restore part of its normal functional contribution by partially restoring vision.

### 2.3. Functional Connectivities Creating Efficiencies of Response: Creating Associative Educational Networks

We possess, especially as adults, a significant degree of localization of function which is inadequate to explain our capability for neuroplasticity, regeneration, spontaneous recovery, and optimization. In childhood more brain area is dedicated to language acquisition which over time concatenates to highly specialized regions for the purposes of optimization [36,37]. Neonates are born with a paucity of connections which, during early and middle childhood exhibits exuberant connectivities allowing for the capacity to make varied permanent specialized connections that will eventually pare down to support efficiencies of response. The histologies are represented in Figure 2.

What we can learn from these changes in connectivities is that the child learner has significant degree of flexibility in making hard-wired connections. If one were to ask an elementary school child to imagine a dolphin bouncing a pineapple on its snout, neurologically it would represent a complex problem requiring sophisticated coordination inside of the child’s brain. The reason being is that in order to create a novel unusual image, the child’s brain would be required to take familiar pieces and assemble them in new ways. That is because in order to create these strange new images the child’s brain would have to take familiar pieces and assemble them in novel ways—like a collage made from fragments of photos. The child’s brain would be required to juggle a sea of thousands of electrical signals—getting them all to their destination at precisely the right time. It turns out that this is actually a complex problem that requires sophisticated coordination inside of our brains. The brain has to negotiate a sea of electrical signals getting them all to their destination at precisely the right time.

When we look at an object, thousands of neurons in our posterior cortex fire. These neurons encode various characteristics of the object—spiky, fruit, brown, green, and yellow. Synchronous firing strengthens the connections between that set of neurons linking them together into what is known as a neuronal ensemble. In this case—the one for pineapple [38].

We can hypothesize and state that in a theory of mental synthesis—timing is key [39,40,41]. If the neuronal ensembles for the dolphin and the pineapple are simultaneously activated, the child could perceive the dolphin and the pineapple simultaneously. Without synchronization, the pineapple and dolphin would be asynchronous—two separate objects but not a single image. Something in the child’s brain has to coordinate that firing. A plausible candidate is the prefrontal cortex—which is involved in all complex cognitive functions. Prefrontal cortical neurons are connected to the posterior cortex by long spindly neural fibers with the prefrontal cortical neurons sending electrical impulses down these fibers to multiple ensembles in the posterior cortex thereby activating them in unison. If the neuronal ensembles are turned on at the same time, then the child would experience the composite image as if he or she had actually seen it. This conscious purposeful synchronization of different neuronal ensembles by the prefrontal cortex is an example of mental synthesis. In order for mental synthesis to work, signals would have to arrive at both neuronal ensembles simultaneously. The problem is that some neurons are much further away from the prefrontal cortex than others. If the signals traveled down both fibers at the same time they would arrive out of sync.

One cannot change the length of the connections but the brain—especially as it develops in childhood—does have a way to change the conduction velocity. Neural fibers are wrapped myelin, Myelin as an insulator can speeds up the electrical signals. Some nerve fibers have as many as one hundred layers of myelin while others only have a few and fibers with thicker levels of myelin can conduct signals as much as 100 times faster or more than those with thinner ones. We have begun to think that this difference in myelination can be the key to uniform conduction time in the brain and consequently to our mental synthesis abilities [42]. A significant amount of this myelination occurs in childhood and so from an early age, our vibrant imaginations may have a lot to do with building a brain who myelinated connections can craft creative symphonies throughout our lifetimes [21,22]. We have theorized about such processes elsewhere cf [43,44].

### 2.4. The Timing of Network Building: Critical and Sensitive Periods in Neurocognitive Development during Preschool and School Years

The normally developing brain has a lawful progression based on cell division and migration, and development of associative networks that besides the genetic imperative can be largely influenced by stimulation extrinsic to the child’s nervous system [45,46,47]. These extrinsic stimuli can significantly influence neuroplasticity that can support the development of a given function. The ability of the brain to reorganize during a specific developmental time window is known as a “critical period”. Developmental experiences have a significant effect on the structure and function of the brain’s associative networks in the developing child [48,49]. According to Dehortrer and Del Pino [48] and others [50,51,52] it is during an optimal period that a relatively enduring alteration occurs that is required for effectively operating brain function to occur. If that experience occurs during an optimal or sensitive period permanent differences can be found distinguishing,, for example musicians from non-musicians and from those trained early in life rather than later [50,51]. As sensitive periods of experience-dependent plasticity may allow for compensatory or alternative patterns of connectivity, it allows for the “tuning” of the execution of a given function by providing flexibility in functional network development. The specific school-based skills associated with critical and sensitive or optimal periods are represented in Figure 3.

Children during early childhood possess significant abilities in adaptation allowing them to acquire knowledge and behaviors that are continually being refined on the basis of experience with a significant capacity at generalization [53,54]. A major function of the classroom in early childhood education in the context of critical periods involves an equilibrium in plasticity-stability between various brain regions that penultimately will lead to skill optimization of cognitive function largely based on sensori-motor stimulation [55,56,57,58].

Neuroplasticity is a crucially important aspect of child development in general and education in particular as it underlies our ability to learn, remember, and adapt to environments and is continually in flux during the early preschool and elementary school [57]. The brain is singularly plastic during early development when neural networks change as a consequence of sensorimotor experiences. Neuroplasticity becomes less important as the child’s physiology becomes more stabilized through what the extensive literature refers to as developmental stages, in which the process of neuroplasticity continues but on a significantly smaller scale [59,60].

**Figure 3 brainsci-12-01501-f003:**
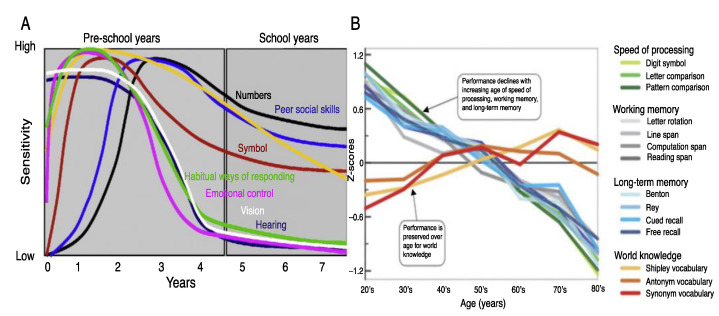
Sensitive periods during early brain development for (**A**) Sensation, emotional control, numerosity and symbolic representation, working and long-term memory and vocabulary. (**B**) Processing speed, memory, and word as a function of age [20,21,48,61,62,63].

The brain’s ability to modify itself as a consequence of experience and environmental allows us to examine the nature of patterns of connectivity and information flow during early childhood development. The fundamental thinking on the subject of synaptic plasticity as a consequence of learning was originally proposed by Hebb in 1949 [64] when he theorized that neurons could drive the activity of another neuron and in so doing, their connection strength is reinforced [44]. Hebbian plasticity indicates that repetitive stimulation of the postsynaptic neuron will be associated with increased synaptic efficacy involving: (a) type of neural signals that initiate change (e.g., inhibitory of excitatory); (b) increased or increased efficacy; (c) the reasons for the synaptic change; (d) the mechanisms involved in their persistence and (e) the time course, be the stimulation transient or longer term [64] with the most elementary form of plasticity. The simplest form of Hebbian plasticity being synaptic strength [43,44].

### 2.5. Timing, Associations, and the Building of Learning Networks

Obviously, there can be no learning without memory. What is learned requires encoding, integrated into memory, and stored for further use. Learning and memory are necessarily linked in the brain. One has only learned something when it is coded in some form of memory for recall or use as a skill in the future. Learning then necessitates the acquisition of information and the strengthening of that information with previously acquired knowledge and its associated networks in memory. Memory is not a unitary process but rather a collective of different processes each with different characteristics. For example, there exists memory for skill, facts, and life events distributed and retrievable differently [35,38]. Knowledge is represented in the brain through interconnectivities of networks. Recall and its associated retrieval processes are associated with spreading activation which then can lead to the recall of associated memories [65]. Our brains then, in Hebbian terms, learn by making and strengthening connections through association and context. Knowledge is thought to be coded in the brain in an interconnected network, with similar or associated items and concepts being more strongly connected [65]. Recall or retrieval of some fact, memory, or piece of knowledge will spread activation to help the recall of other similar and associated memories or knowledge. These connections between neurons through the synapses are continually changing with learning and experience. Learning in the brain is the process of building and strengthening synaptic and regional connections and connectivities [66,67].

In 2005 a group of researchers [68] recorded activity from single neurons in the brains of individuals undergoing surgical intervention for the treatment of intractable seizures. These investigators found the existence of “Jennifer Aniston cells” or neurons that fired when viewing pictures of the actress Jennifer Aniston! This cellular network did not respond when stimulated by pictures of other familiar people, buildings, or objects. When researchers find a neuron firing to a particular concept, this neuron is part of a cellular network. The network of cells in question specifically fired their signal, communicating to the rest of the brain, only for Jennifer Aniston. This is consistent with the notion that specific concepts, are coded by distinctive activity patterns in different sets of neurons. This argument, while part of the formidable questions that have arisen in developmental cognitive neuroscience is complex and nuanced and requires a copious review of the available literature cf. [69,70,71,72,73,74].

## 3. Facilitating Neuronal Connections in Classroom Learning

### 3.1. The Importance of Connections: Learning by Association and Context

As a result of learning and experience, memories, knowledge bits, and experience are encoded in our brains in unique neurons and connected by associational and interconnected networks and cell assemblies. These networks are constantly capable of change. A thought or experience such as the smell of a particular flower can lead to spreading activation that would link to associated concepts already in memory. Our associative networks coded memory and that, in part, is a function of dreaming.

If one were to ask a child to name the capital of Italy, he or she might think of green, white and red vertical stripes, spaghetti, pastas, pizza, and hopefully also Italian Opera, Vivaldi and Verdi. Activation will broaden to all things that the child has individually learned that in turn are connected with Italy. The result is easier or automatic recall of knowledge bits and associated memories and this all as a consequence of Hebbian-type learning who.se function is to strengthen learned material.

For example, if one encounters the smell of lavender and your grandmother at the same time, your lavender cells and your grandmother cells will form a connection. That connection will be strengthened the more often you encounter those two together. In the future, when the two share a strong connection, activation of one will lead to automatic or easier activation of the other. In psychology, the same idea is known as *associative learning*, whereby items that are commonly encountered together lead to stronger association so that recall of one will lead to automatic or easier recall of the other in the future. This process of making and strengthening connections between neurons that are encountered and activated together is thought to represent most learning in the brain.

In early childhood education, this concept becomes relevant in conveying simple to complex knowledge. If one asks a preschool child to verbally describe a cat, they may say something like “a small *animal* with fur that says meow”. This would demonstrate that the child’s concept of cat is encoded by features that the child had previously associated and integrated (animal, meow fur, etc.). Similarly, if one were to ask a cognitive science professor a question about memory consolidation he or she might be likely to reply in terms of sleep, infants, neonates sleeping approximately eighty percent of a twenty-four hour cycle ultimately tapering off to about fifty percent after six months and associated with the redundancy of information in the context of limited capacity for acquiring information, work on the consolidation of memory during sleep, rats learning to run a maze, forgetting how to do so after electrical shock to the temples.

Knowledge is therefore thought to be coded in the brain by unique patterns of activity across different sets of neurons that are linked and spread activation to other associated concepts, memories, or pieces of knowledge. Learning involves making or strengthening the connections between concepts as they are encountered or recalled together. That is the function of childhood education.

This concept is most readily seen when examining the nature of embodied language., It is less important to understand that Broca’s area is in the frontal lobes, anterior to the Sylvian fissure is responsible for expressive language, and that the temporal lobes, and Wernicke’s area, in particular, in the left temporal region is responsible for receptive language. More to the point, as evidenced in Figure 4, is that the networks are different for leg, face, or arm rerated words [75]. Education’s function is to build associative networks of the kind evidenced in embodied language [76,77].

The function of childhood neurological development is precisely to facilitate the creation of localized function and it is dynamic interactions with other brain regions forming associational networks or ensembles. It can be changed and is therefore plastic. This localization of function is not the explanation of a process, but rather the end-result of training. The efficiency of cognitive function is directly a consequence of the effectiveness of networks that now can be measured [78]. Fewer brain regions necessary to accomplish a single task in one individual compared to another for the same task is a measure of efficiency. It is an example actually of why developmental cognitive neuroscience needs to, among other things, address classroom learning and developmental issues. Mathematics has been involved in this issue for hundreds if not thousands of years. Such a metric and its application has not been particularly applied in the behavioral sciences and education with perhaps he exception of Human Factors Psychology. In mathematics, we can examine path length to construct the shortest path [78]. A shortest path one that exists between two nodes with the least number of edges if the cost (a measure of how much effort it takes to travel along an edge in a network) of traveling along each edge is the same. Actually, given origin and destination nodes, a shortest path from the origin to destination nodes is a path with the lowest total cost among all paths from the origin to the destination. So, the young lady with the congenital hydrocephalus in Figure 1 is a more efficient processor than a young school aged child (Figure 2) and an adult brain has more efficient processing than that of a school aged child as there is a relative paucity of connections in the adult compared with the young child. In other words, pruning, by this standard, makes the brain more efficient (or optimized in production management terms). How this applies in education remains to be seen.

These networks, active during learning and problem solving of all kinds, are plastic and can be changed as a direct consequence of experience and training. In attempting to apply graph theory concepts to child and adolescent neurocognitive performance to create a fundamental change in the educational training and evaluation paradigm, we can characterize the organization & development of large-scale brain networks as represented in Figure 4 [21,22,66].

One can conclude up to this point in the discussion that rote learning and memorization of unrelated facts is not efficient as it is, generally speaking, not the way the brain forms associative networks for effective learning. Actually, the brain learns by forming and strengthening connections between associated concepts so that learning in context and linking and forming associations with allied knowledge will be more effective for retention. This is seen very clearly in the benefit of using mnemonics that can greatly help learning by creating meaningful associations between elements. Using the phrase “’i’ before ‘e’ except after ‘c’” is an example of a mnemonic to remember the correct order of the ‘i’ and the ‘e’ in the word ‘friend’.

Activities that promote recall or reevaluation of knowledge are highly beneficial for learning by reinforcing and further strengthening the connections between associated items that are recalled together. Examination and assessment should therefore be structured in a way that aids the recall of associated concepts, reinforcing the neural connections between these items and further consolidating learning. In this way, appropriately structured examination can be an integral part of learning itself.

### 3.2. Physical Activity and Cognitive Relationships

Brain-Derived Neurotrophic-Factor (BDNF) increases after exercise promoting neuronal size, dendrite branching, and spine number in the rodent hippocampus as is evidenced in Figure 5.

More importantly, physical activity directly relates to the ability to significantly improve in reading, mathematics instruction, spelling an academic performance in general. With complex movement integrated spatially greater brain areas. Figure 6 illustrates the relation between physical activity and interregional connectivities in the brain for learning.

### 3.3. Implications of the Development and Plasticity of the Neural Connectivities for Learning and Instruction

#### 3.3.1. State of Our Understanding of Brain-Based Learning

We have already noted that neuroplasticity is simply the ability of the brain to change as a result of learning. There exist at least two types of modifications occur in the brain with learning: (A) A change in the internal structure of the neurons, the most notable being in the area of synapses and (B) an increase in the number of synapses (more dendrites) between neurons. The function of education is simply put to grow dendritic connections. Growing dendrites (size and numbers) physically increases brain weight with new growth on frequently used neurons [81].

What we have learned about effective learning is that (A) introducing and reviewing information in several ways creates synaptic connections [45] (B) The more connections are utilized the stronger the connectivities become [21,82]. (C) The more complex skills required in a given occupation or task the more dendrites are found on relevant neuronal connectivities [83]. (D) The branching of nerve cells occurs primarily at night during various times during our sleep cycle. Sleeping after learning grows twice as many neural dendrites as when learning the material, with the bulk of hard wiring takes place during sleep [84]. Children who are sleep deprived after learning new information are unable to process and use the information as well as children who are not sleep deprived [85] (E) Dendritic connections grow from what exists [86]. (F) Dendritic connections grow from what is practiced [87,88]. (G) Dendritic growth is associated with novel stimulating and attention provoking experiences [89]. (H) Dendritic connections affected by emotion and learning [90]. (I) Use it or lose it! Ineffective or weak connections are “pruned” in much the same way a gardener would prune a tree or bush, giving the plant the desired shape [91]. (J) It is neuroplasticity that enables the process of developing and pruning connections, which, in turn, allowing the brain to adapt itself to its environment [92,93]. (K) The density of synapses declines in association with selective pruning of redundant or unused connections [93]. (L) Synapse and therefore connection formation continues despite ongoing pruning [93]. (M) Once believed that with aging, brain’s networks become fixed, but we now know otherwise. In past two decades, research has revealed that brain continually changes and adjusts over the lifespan [94].

#### 3.3.2. More Comprehensive Understanding of the Power behind the Spinal Cord and Lower Parts of the Brain, Especially the Cerebellum in Classroom Learning

The cerebellum contains more neurons than the rest of the brain together [95]. It coordinates movement and is important in performing timing of complex motor tasks as indicated in Section 6 above (e.g., smooth a dancer’s steps) [22]. The cerebellum stores memory of automated movements (e.g., shoe tying, typing) improves performance for movement sequencing of greater speed, accuracy, and less effort (violin playing) [58]. The cerebellum is also involved in mental rehearsal of motor tasks which to improve performance and make those tasks more skilled [22].

It had previously been thought that the cerebellum’s function was exclusively related to motor control, but recent evidence has demonstrated otherwise with some reports having also indicated relationships with non-motor cortical functions [96,97,98]. Recent evidence has indicated da relationship between the cerebellum and functions such as attention [99,100], language [100,101]. as well as mental imagery [102,103]. In actuality, estimates based on fMRI studies have indicated that a significant portion of the cerebellar cortex is linked to cortical association areas [104].

We can best understand the function of the cerebellum in the context of learning behavior less by its behavioral effects but more by its computational capacity. In a way similar the functional organization of the neocortex, the cerebellum consists of numerous independent segments that had been assumed to be performing a single computation. However, if both input and output connectivities are functionally connected to motor regions then that function will be associated with motor behavior and motor learning. Alternatively, if the connectivities functional project to neocortical regions then the function will relate to non-motor cognition. We therefore can understand the role of the cerebellum better by examining its relation to behavior and affect [105,106,107,108].

Doya [96] had suggested the function of the cerebellum could be best be modeled as a “device” involved in supervised learning which can be exemplified as an algorithm that can seek and identify categories for unseen instances or developing associational networks to generalize from the training to unseen situations. The basal ganglia, in contradistinction functions to support reinforcement learning exemplified by the lack of a requirement for input/output but in finding a not needing labelled input/output pairs be presented, and in not needing sub-optimal actions to be explicitly corrected. Instead, the focus is on finding balance between exploration of some novel presentation and current knowledge. The neocortex, in Doya’s [96] model is the part of the learning system involved in *unsupervised learning* or probabilistic methods involved in mimicry where the result is the building internal representations of knowledge that can then generate imaginative content. Supervised learning, in contradistinction, requires the labelling of objects and thereby classification by experts or teachers as opposed to unsupervised learning which is built by self-organization on the basis of probabilistic models [96]. It is in this understanding that we can model creative intuition of the “ah ha” moments in working memory [109].

The spinal cord & lower parts of brain perform skills automatically, without conscious attention to detail. This then allows the conscious brain to attend to other cognitive task with the benefit of facilitating the learning process by physical activity. We are quite capable of walking, talking, and thinking simultaneously with walking facilitating memory. The same holds true for driving a car and thinking.

## 4. Neuronal Systems Known to Facilitate Classroom Learning

### 4.1. Attention

Before individuals can learn or have a skill rehabilitated, in most instances but not always, something or someone, must capture their attention. The most effective ways to gain attention is through: (A) Novelty [110], (B) Humor [111,112] and (C) Surprise [113,114]. The brain is a novelty seeker (changes in environment; something new or different). The brain is always looking for stimuli by attending to relevant stimuli. A system in the lower brain (reticular activating system) filters stimuli and decides what to attend to and what to ignore based on physical need, novelty, and self-made choice [115]. Focus depends on relevance and meaning [6,116]. The brain attempts to make sense of its world and determine if information is meaningful. For information to be meaningful the learner must care about the information or consider it important. Even if individuals understand what activity is being required, if the information is not relevant and does not connect to their past experiences, it is unlikely to be recalled. To make information meaningful the educator must employ past experiences on which to “hook” onto the new information and create new experiences with them. For a more expansive understanding please refer to our published work on the topic [22].

### 4.2. Emotional Hooks in Learning

Emotional responsivity, thinking and learning need to be linked cf. [117,118] and these linkages can be evoked in at least two distinct ways: (A) By recreating the emotional climate of the environment in which the learning originally occurred as emotions are associated with the learning content [119]. (B) Emotions drive attention, create meaning, and have their own distinctive memory pathways [120,121]. Emotions are not located in a single “emotional center”, but are interactively distributed throughout the brain [122]. This being the case, then the limbic system must be stimulated to create support for effective learning [123]. Table 1 outlines the connections of emotional responsivity with the learning process.

### 4.3. Complexity v. Difficulty

In examining teaching materials that are designed to advance a learner’s thinking and skill sets in various content areas, the natural tendency for teachers is to increase task difficulty rather than complexity as the challenge mode [124]. Complexity is the thought processes the brain uses to deal with information in comparison to difficulty which refers to the degrees of effort that the learner needs to expend within a given level of complexity [125,126,127]. Higher order thinking increases understanding and retention [127,128,129,130]. Our ability to learn, remember, and recall depends on number of connections between neurons. PET scans show that elaborative rehearsal with higher-order thinking skills, engages the brain’s frontal lobe [6,131,132]. This helps ALL learners to make connections between past and new learning, create new pathways, strengthen existing pathways, and increase the likelihood that the new learning will be consolidated and stored for future retrieval.

### 4.4. Creativity: A Principal Goal of Instruction Is to Be Able to Use Knowledge in Different Settings

When deciding on how to use rehearsal teachers need to consider the time available as well as the type of rehearsal appropriate for the specific learning objective. Rehearsal does not guarantee transfer to long-term memory [133,134]. However, almost no long-term retention can happen without rehearsal and sleep solidifies it [134]. The essential differences are described in Table 2.

### 4.5. Memory Considerations in School-Based Learning

School teachers with a grounding in developmental cognitive neuroscience could possess a greater understanding of the types of memory and how they are formed can select strategies that are more likely to improve the retention and retrieval of learning as represented in Figure 7. Learning and retention are different. We can learn something for just a few minutes and then lose it forever. Retention can be affected by many factors that include: focus effectiveness of individual [135], rehearsal length and type [136] identification of the critical attributes of the information being taught [137], the learning pattern of the individual [138], and prior learning’s inescapable influence on the effects of pro- v. retroactive inhibition [139,140].

In classrooms, educators largely teach to semantic memory, when students are sitting in classrooms [141,142,143]. It is a less optimized form of instruction since there exists less control over the input in the nervous system [144,145]. It is easier to teach to episodic memory (hands on or learning experientially) [146]. Individuals would learn and remember whether they wanted to or not. The learning is then under the instructor’s control. By adding a hook to an emotional component and the likelihood for retention will be significantly increased [114]. Flashbulb memories, for example, are episodic memories with an emotional attachment.

## 5. Discussion

In providing some direct examples of how developmental cognitive neurosciences can readily be applied to the classroom based on what we have described here, we can note that cognitive tutoring may be effective in providing innovative and practical software for everyday use [147]. Separately, we know that sleep patterns change as a function of development [148]. Adolescents tend to sleep longer than do other age groups consequently affecting early morning cognitive capacity. Additionally, the circadian rhythms of teenagers are different when compared to other age groups. School and class scheduling needs to account for that. Although repetition of material is necessary for effective learning [20] the cognitive neuroscience literature has noted the importance of a “spacing effect” where we have learned that students retain significantly more material when learning sessions are spread over time as opposed to being provided in a single session [149]. The consequence is to provide variety in the classroom with material reiterated over the course of a semester rather than in a session or in a few days. We know that each of us possess similar brain anatomical structures, but the neural ensembles associated with that anatomy function in a highly individualized manner [3,150]. Therefore, learning tools that are adjustable to the needs of an individual learner are particularly valuable in the classroom. We have noted that if “you do not use it you lose it”. However, we know that individuals who read more challenging books possess a significantly greater variety and number of neural connections. This research can be practically applied to the classroom in obvious ways [151].

Other examples of translational neuroscience for classroom applications would include the relevance of making learning a positive experience which is in turn related to the release of dopamine, which in turn aids in memory for facts [152]. The relation between movement and cognition associated with the release of brain derived neurotrophic factor (BDNF), associated with neuroplasticity and associational networks, discussed in greater detail in the paper, is yet another example of how the marriage of developmental cognitive neurosciences and education may be applied [153].

Education researchers are beginning to approach complicated functions such as memory and learning physiologically [154,155]. We think that the knowledgebase of the neurosciences, human factors and even industrial engineering that address, among other things, notions of efficiency, can offer benefit to education throughout a learner’s life [21]. Recent research on the human brain is providing us with a better understanding of the processes of human learning in the classroom and thereby improving teaching methods [21,155,156,157,158,159].

Not infrequently when an initial result is reported in the cognitive neuroscience literature it is seized and expanded upon with not much thought given to the fundamental nature of biological process. With varying degree of success but mostly not, over the last 20 years, education has been examining and attempting to adapt findings from the neurosciences to better inform education practice and policy. A good example is that of the decision in 1998 of the State of Georgia to fund an expensive program, to provide CDs of Mozart’s music to all new mothers [160]. The Governor of Georgia, in creating this policy, based his decision on reports from the literature of cognitive neuroscience performed at the University of California, Irvine [161]. The governor thought that the educational system in Georgia would be able to “harness the ‘Mozart effect’” for Georgia’s newborn infants by introducing classical music to facilitate brain development. Unfortunately, when examined more closely, the research on the so called “Mozart Effect” had not much to offer education. One study, reported in Nature [162], found that listening to Mozart’s music raised the IQs of college students for a brief period of time. Another study found that keyboard music lessons boosted the spatial skills of three-year-olds [163]. Cognitive neuroscientists were puzzled by the program in Georgia that was based on their work. Since this fiasco, numerous researchers have emphasized caution in the interpretation of findings without significant laboratory support to implement practical applications of potential educational interventions [164,165,166].

Even worse than the example cited above are government-ordered educational mandates. In 2010, for example, the US State of California had greater than 50 such mandates that required specific activities to be performed by school districts at a collective cost then of over 400 million dollars. Many of these requirements did not benefit students or educators. The existing mandate system also can reward districts for performing activities not only inefficiently but ineffectively [167].

Most currently prevailing approaches to teaching possess a misunderstood, inappropriate and significant bias to the functioning of the left hemisphere [168]. Reading, writing and arithmetic are all linear and rational processes. Most instructional approaches have tended to prolong and aggravate this bias. There is tendency to diminish imagination and fantasy, visualization, and inference, in the interests of rote-learning, reading, writing, and arithmetic. There exists an educational milieu that emphasizes verbal skills, “Using words to talk about words that refer to still other words” [169].

School systems seem to have a predilection to reflect our western culture in which left hemisphere skills are favored over right hemisphere activities [170,171]. We emphasize in our society “propositionality” at the cost of “appositionality” [172], resulting in both adjustment difficulties as well as in skewed education and training [173]. Our students are being offered the education that does not provide them with an understanding of the complex nature of the world and themselves, an education for the whole brain [82,174]. Sperry wrote: “Our education system and modern society generally (with its very heavy emphasis on communication and on early training in the three R’s) discriminates against one whole half of the brain. I refer, of course, to the nonverbal, non-mathematical, ‘minor’ hemisphere, which we find has its own perceptual, mechanical and spatial mode of apprehension and reasoning. In our present school system, the attention given to the minor hemisphere of the brain is minimal compared with training lavished on the left, or major hemisphere” [175].

Educational institutions have placed a great value on verbal/numerical skills and categorization and have methodically disregarded those proficiencies that would support young children’s development of imagination, visualization, and/or sensory/perceptual abilities [176,177,178]. Dylan Williams [179,180] indicated that in his review of the literature, practiced and trained skills do not typically generalize as being significantly related to specific training. He stated, “*It is certainly unhelpful, and probably wrong, therefore, to talk about ‘critical thinking skills’. Critical thinking is an important part of most disciplines, and if you ask disciplinary experts to describe what they mean by critical thinking, you may well find considerable similarities in the responses of mathematicians and historians. The temptation is then to think that they are describing the same thing, but they are not. The same is true for creativity. Creativity is not a single thing, but in fact a whole collection of similar, but different, processes. Creativity in mathematics is not the same as creativity in visual art. If a student decides to be creative in mathematics by deciding that 2 + 2 = 3, that is not being creative, it is just silly since the student is no longer doing mathematics… Creativity involves being at the edge of a field but still being within it*”.

The overly reasoned representations so frequently offered to children in their textbooks highlight linear thinking and discourage intuitivist, metaphorical and analogical thought. These elements of neural performance among children have been left to adaptation by unplanned environmental rather than systematic curriculum design. Education, which is largely conceptual, verbal and reading-based generally does not provide for concrete, esthetic experience, especially with subjective internal operations [181,182,183]. Education’s structure levies didactic instruction, or a logical-objective dominance over the subjective-intuitive, and right-wrong criteria, quite early in the course of emergent awareness of the child’s world of him or herself. Except in atypical cases, creative potential is inhibited or diminished [22,184,185]. This leads us to conclude that typical western education systems support underdevelopment of the right hemisphere. We draw these conclusions as a consequence of the prominence of verbalizing, intellectualizing, and conceptualizing ‘curricula’ which has become equated with understanding and less so with creative thinking and imagination.

This imposes “neurotogenic limitation” and fixes cognitive processing so strictly that they inhibit the ability to integrate new information. According to Gazzaniga [186], school curricula are, “Inordinately skewed to reward only one part of the human brain leaving half an individual’s potential unschooled”. Traditionally, education has been preoccupied with the “formal discipline” which effectively blocks the learner from recognizing and cultivating creativity. Educators have attempted to integrate the cognitive neurosciences into the classroom by placing undue reliance on functional neurological models that may be inappropriate and damaging to the performance of children in classroom settings, both instructionally as well as in evaluating learning performance.

## 6. Conclusions

The brain continuously remodels itself-even into adulthood. Synapses (dendrite growth) continue to be formed in the brain. Lifelong enrichment experiences are important for continued dendrite growth and healthy functioning brains. If one does not use it one will lose it—therefore repetition becomes critically important. The brain is “pattern seeking”, likely as a result for the necessity for building associative networks supporting the neuroplasticity that in turn relates to the generation of BDNF and other proteins supporting dendritic connections and neuroplastic processes in development. The brain seeks to make order out of chaos and therefore there is a continual search for meaning and pattern detection that should be supported by “mind map” and graphic formats.

We have endeavored to provide an overview of the efficiencies of neural processing as a basis to the development oof alternative approaches and thinking to classroom learning, teaching, facilitating creative thinking, curriculum design, and subsequently to policy and leadership informed by current understandings in the developmental cognitive neurosciences and by optimization principles applied to schools, and learners.

We have noted that brain connectivities are variously organized efficiently or inefficiently in systems that can be relatively easily measured. The optimization of changes in brain activity associated with training and learning can be relatively easily evaluated. In some learners, delayed or different mechanisms of brain connectivity change can be examined as a consequence of instruction and experience. These changes will be most certainly associated with functional connectivities in the brain.

Grade level-based skill and function measurement or other binary considerations of whether a learner does or does not possess a given skill, “medicalizes” the paradigm of learning. The focus should be less on binary thinking and more on optimized performance and learning strategy and associative networking, most easily measured by brain-based considered and strategic solutions. For example, individuals who are late learners of a second language demonstrate brain activity in brain areas that are not optimally synchronized and coordinated. With continuing brain development, more simultaneously active but distant regions require synchronization for mental synthesis. It is the developmental lack of effective synchrony that we hypothesize addressing the effectiveness of the connections between cognitive and motor functions and can address the very nature of learning itself. We have the tools to make the learning process more efficient. They simply as yet not been implemented effectively in childhood education

## Figures and Tables

**Figure 1 brainsci-12-01501-f001:**
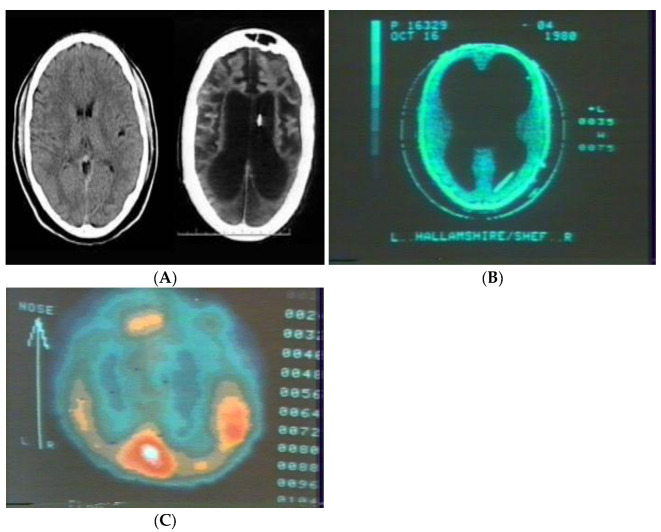
(**A**) Normal CT-scan (Left) and that of patient T.S. in persistent vegetative state resulting from anoxia during childbirth (Right). (**B**) CT of congenitally hydrocephalic individual within the normal to above average range of intellectual capacity. (**C**) Xe-gas regional cerebral blood flow of individual in (**B**) while performing mental arithmetic.

**Figure 2 brainsci-12-01501-f002:**
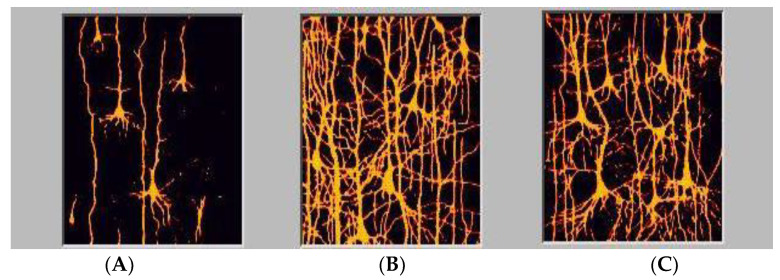
We see the complexity of dendritic structures of cortical neurons consistent with expansion of synaptic connectivities as a function of age. ((**A**), 5 days; (**B**), 6 years, (**C**), adult) [21].

**Figure 4 brainsci-12-01501-f004:**
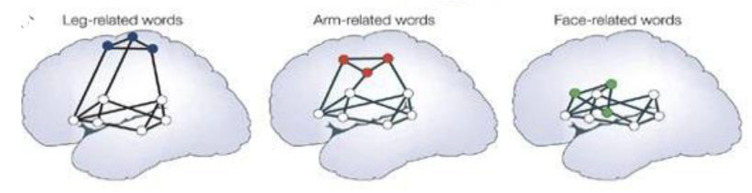
Multiple stream models of receptive language functions organized into multiple self-organizing simultaneously active networks. Grounded meaning indicates that the meaning of words and sentences are “embodied” [21,66].

**Figure 5 brainsci-12-01501-f005:**
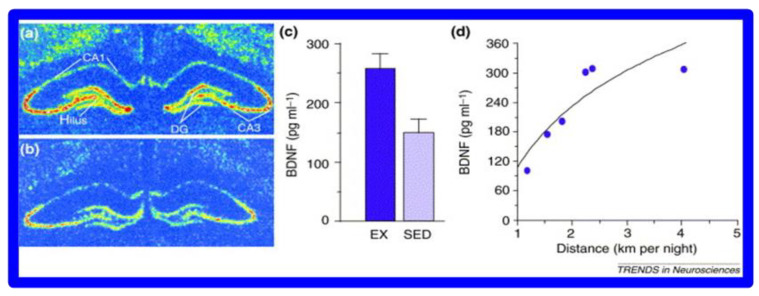
BDNF increases in mouse hippocampus (**a**) After 7 days of volunteer wheel running compared to (**b**) sedentary (SED) mice. (**c**) BDNF levels as a function of exercise (EX) v. sedentary behavior (SED) (**d**) correlation between the distance of wheel running and BDNF levels. (with permission [79]).

**Figure 6 brainsci-12-01501-f006:**
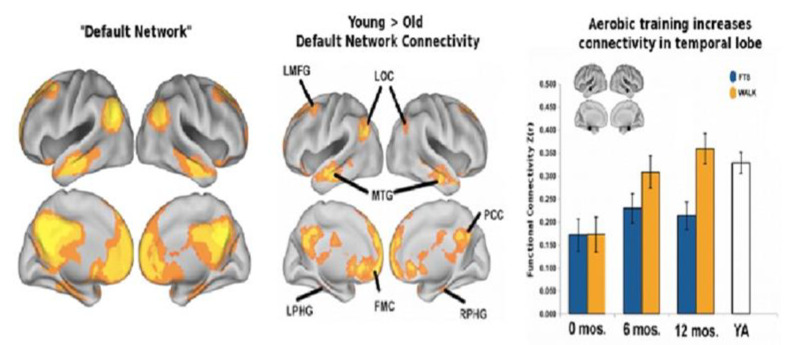
The left-hand panel represents the “default network” (DN) in healthy students of college age. The aging process has negative effects on network interconnectivities represented in the middle panel in particular within a major regional hub that involves the posterior cingulate. In contradistinction, the righthand panel inter-region connectivities (synchronous activity) of the DN following a one-year of aerobic exercise program in comparison to the non-aerobic control group. Walkers after a year demonstrated interregional connectivities not significantly different than college aged young adults (YA) (with permission [80]).

**Figure 7 brainsci-12-01501-f007:**
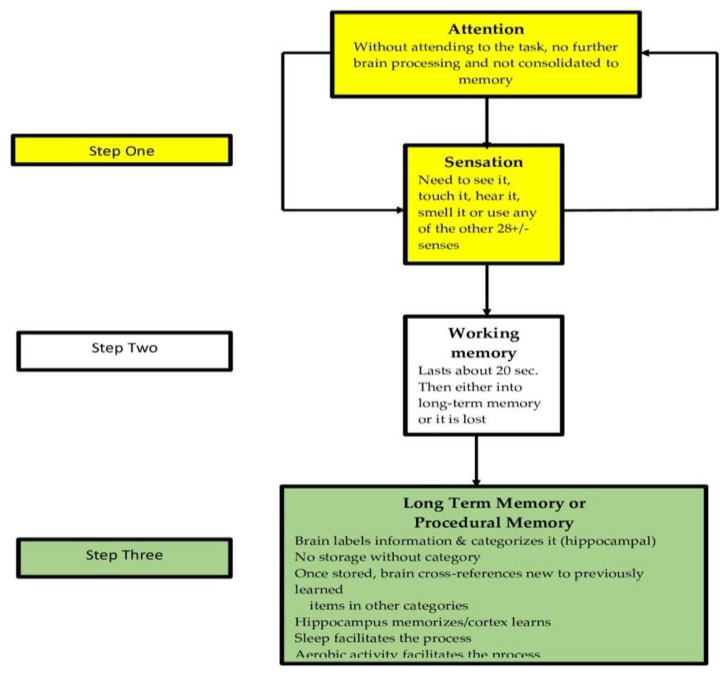
The key to teaching is to help students put things into memory and then retrieve the information.

**Table 1 brainsci-12-01501-t001:** The Requirement for the integration of meaning, emotions, and learning.

Meaning, Emotions and Learning	However, When Individuals Feel Helpless and Anxious…	However, There Must Be Some “Challenge”
➢ New learning is more likely to be attended and remembered if it has meaning and contains emotional “hooks”. ➢ Optimal emotion level is necessary for learning to take place.	➢ Amygdala becomes overactivated, new information prevented from passing to memory. ➢ Information must get through amygdala to get to hippocampus for memory storage and reasoning parts of the brain.	➢ Must have mild to moderate challenge to stimulate authentic curiosity and engagement in lessons. ➢ This will motivate students to work toward greater understanding and connection with the material.

**Table 2 brainsci-12-01501-t002:** On the differences between learning memory and retention.

Learning	Memory	Retention
How our brain acquires new information and skills	How and where our brain stores learned information and skills	How long-term memory preserves learning so that it can locate, identify, and retrieve it accurately

## Data Availability

Not applicable.

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
