# Peer review of "On the Application of Developmental Cognitive Neuroscience in Educational Environments"

_brainsci, 2022, doi:10.3390/brainsci12111501_

Round 1

Reviewer 1 Report (Previous Reviewer 2)

Αs already stated this is an interesting article.

The author made the necessary corrections.

Thank you

Author Response

Nothing to add and a thorough spell check will be performed.

Reviewer 2 Report (Previous Reviewer 1)

The author has clearly made some amendments to this manuscript and taken on board some advice from early round of review. They should certainly be commended for this. Unfortunately I still find it difficult to understand the narrative presented in this work. The aims of the manuscript are not clear and I find it very difficult to follow the argument presented in the prose. I think there’s a really interesting argument to be made that neuroscience could do more than just tweak current educational practice, but could really overhaul it by demonstrating the value of creativity. I very much hope that the author is able to re-work this piece, find that argument and present it in a digestible way. Such a piece could have a powerful impact on education policy by presenting evidence that is already available and suggesting the kind of evidence that would be pursuasive so as to guide collaborations in education and neuroscience.

Author Response

The author has clearly made some amendments to this manuscript and taken on board some advice from early round of review. They should certainly be commended for this. Unfortunately, I still find it difficult to understand the narrative presented in this work. The aims of the manuscript are not clear and I find it very difficult to follow the argument presented in the prose. I think there’s a really interesting argument to be made that neuroscience could do more than just tweak current educational practice, but could really overhaul it by demonstrating the value of creativity. I very much hope that the author is able to re-work this piece, find that argument and present it in a digestible way. Such a piece could have a powerful impact on education policy by presenting evidence that is already available and suggesting the kind of evidence that would be pursuasive so as to guide collaborations in education and neuroscience.

I am glad that the reviewer sees the value of the ms., and has no intrinsic problem with the argument presented at this point especially after the "teeth grinding" last round for which I am much appreciative. Essentially the manuscript has been reorganised with the aims clearly stated and this I have endeavoured to do - hopefully. In particular, the introduction has been significantly cut and many of the opening arguments have been reassigned as part of a didscussion. In addition, and whilst not characteristic for Brain Sciences journal, I have included an outline of the entire paper prior to the Introduction.

Reviewer 3 Report (New Reviewer)

Dear authors,

Thank you for the opportunity to read this ms. 

Author Response

No response is needed and obviously content with that.

This manuscript is a resubmission of an earlier submission. The following is a list of the peer review reports and author responses from that submission.

Round 1

Reviewer 1 Report

This paper attempts to propose some valuable ideas pertinent to the field of educational neuroscience. In its current form it doesn’t quite accomplish its goals, but with the recommended adjustments I believe it could be effective.

This is an ambitious paper in its scope. It tries to bring together many ideas. As a consequence, the logical argument gets a bit lost. Summarizing the work is not straightforward, but it seems to be a review of neurophysiological factors that could/do influence learning in the classroom.

Based on the title and the early sections of the paper, the author proposes the following thesis: education, as a field, is looking for answers, and in doing so has turned to neuroscience, but the questions being asked are about what areas of the brain are for and how can we apply findings from neuroanatomy to better teach ‘left brain’ functions. The author suggests that what education should be asking is what neuroscience knows about training cortical efficiency and how that knowledge could be applied to teaching ‘right brain’ functions. If more fully explored, that could be an interesting line of thought.

I have made suggestions on the manuscript and would additionally suggest the following changes:

·       Open with a clear statement about the aims of the paper.

·       Structure the paper using a logical argument that moves towards the stated aims.

·       Apply a rigorous edit.

·       Be more awareness of the need to cite evidence in support of key ideas.

·       End with a clear statement in summary, drawing practical conclusions.

I wish the author all the best with this thought-provoking work.

Author Response

What an incredible review! I thank the reviewer for this review and copious comments made throughout the .pdf version of the text. In the interests of clarity, I have addressed the comments provided in the online review system in seriatim. I have separately provided comments about changes and corrections recommended by the reviewer both in the .pdf and .docx versions as well as here in the review forum. The marked and amended .pdf and .docx versions are attached to the comments section here in the review forum.

One additional note hopefully to provide some context. The instant paper is meant as an introduction to a special issue of Brain Sciences of the same title, "The Brain Goes to School" and is meant to tie together the many "loose ends" of the numerous papers to follow. The broad coverage of the review paper then is relevant in that context and the reviewer's comments and suggestions for improvement become even more relevant.

....The author suggests that what education should be asking is what neuroscience knows about training cortical efficiency and how that knowledge could be applied to teaching ‘right brain’ functions. If more fully explored, that could be an interesting line of thought. I have made suggestions on the manuscript and would additionally suggest the following changes:

  • Open with a clear statement about the aims of the paper.

Such a statement has been provided and can be found between lines 24-27 in the revised manuscript and written thusly, "Cognitive Neuroscience and Human Factors Engineering have made many strides over the past eighty years and the application to classroom-based instruction offers the possibility of a fundamental sea change in how the educational product is delivered and acquired. This paper overviews the need for the Special Issue of the same title."

  • Structure the paper using a logical argument that moves towards the stated aims.

The paper has had its organizational structure revised to provide a better flow of ideas as suggested by the reviewer.

  • Apply a rigorous edit.

That has been done line by line and using Grammarly as well. Content has been reevaluated as well

  • Be more awareness of the need to cite evidence in support of key ideas.

That goes without saying and after a careful reread, missing reference support has been added when indicated

  • End with a clear statement in summary, drawing practical conclusions

The suggestion has been implemented.

Comment 1:

Claims as specific as the ones in this paragraph need evidence to support them. This section would be better replaced with a general statement about what this paper aims to achieve.

A number of references have been added to support the contention and labeled now as references 1 and 2. In particular:

Ansari D, Coch D. Bridges over troubled waters: Education and cognitive neuroscience. Trends in cognitive sciences. 2006 Apr 1;10(4):146-51.

Sigman M, Peña M, Goldin AP, Ribeiro S. Neuroscience and education: prime time to build the bridge. Nature neuroscience. 2014 Apr;17(4):497-502.

Additionally the opening statement I think now clearly states the purpose of the paper.

Comment 2:

I think this sentence could be substantially simplified. Some examples would bring this point to life for the reader. 

The sentence has been changed and detailed examples are provided throughout the paper, A general statement has been inserted here.

Comment 3:

Has this not been the case for a while? ("Education researchers are beginning to approach complicated functions such as memory and learning physiologically

More fully explained now, but actually not in my opinion, as there is a disconnect between the resolution in the laboratory in msec or secs and what is required for a better understanding of classroom instruction resolved mainly in days, weeks, months, and years. The exception may be the work on individuals with nonmusical backgrounds who are taught a song played bimanually and changes observed in the locus coeruleus. These changes are not maintained, but after iterations over many months, the results are sustained. (Notably referencing Schlaug’s work at Harvard along with Pascual-Leone) (e.g. Pascual‐Leone A. The brain that plays music and is changed by it. Annals of the New York Academy of Sciences. 2001 Jun;930(1):315-29 Or Wan CY, Schlaug G. Music making as a tool for promoting brain plasticity across the life span. The Neuroscientist. 2010 Oct;16(5):566-77.)

Comment 4:

What do you mean here by 'human factors' and 'industrial engineering'?

Human Factors Psychology, Human Factors Engineering, and Production Management (all subsets of Industrial Engineering) are, among other things, concerned with scheduling, optimization, efficiency, and other matters related to getting a product or service effectively delivered. The tool skills of these disciplines have largely not been employed either in Education or in brain sciences. If one were to understand that educational systems are delivering a product, then the tools of product delivery ought to be employed to maximize and optimize the delivery of those services and evaluate the processes whereby those services are delivered. That includes evaluating brain networks associated with learning. This way of thinking as well as the measurements employed in these fields have neither been adopted nor employed in educational settings nor in any debate on education change, to the best of my knowledge. To include such a long-winded description in this paper I think would both obfuscate the main points and not be quite relevant to the main arguments. However, I am open to adding a bit about that.

Comment 5:

If you can give some specific examples of actual translation to the classroom that would be brilliant!

I have provided some specific examples and have expanded the text significantly in doing so. (See lines 38-108). It is obviously not exhaustive but more of a tease with details in some instances provided further along.

The text now reads:

"We require the advancement of innovative models to better understand activities that can importantly affect motivation, learning, and memory as well as evaluation methodologies that can observe, study and assess these functions. We are slowly realizing that there exists a significant intersection between the problems of psychological, sociological, and educational processes and those of neurobiology, biochemistry, and neurophysiology, with the possibility of reciprocal assistance. [4,5,] [1,2] Neuroscience has influenced school-based activity in various ways. For example, it has provided us with a better understanding of the nature of dyslexia [4] and has offered [5] and insights into how diverse variables such as attention, sleep, relationships and anxiety can affect educational outcomes. [6,7,] Many difficulties exist in actualizing cognitive neuroscience applied to the classroom.  Principally the various disciplines have different end goals such as prescriptive v. descriptive or fact v. solution oriented. Additionally, the neurosciences have been measuring effects in milliseconds to minutes whereas education has been more concerned with changes measured in days, weeks, and years.[8,9]

With this said, there does exist a knowledge base in the cognitive neurosciences that can justify its translation into practical classroom solutions [1,2,3,6,7,8] and this we will provide in greater detail later in the paper. As examples, we know that: a) Teachers do not convey knowledge to learners as learning dynamically builds and rebuilds neural networks [6]; b) a learner’s performance is highly dependent on the context in which the information is provided [10] and therefore, c) learners require the ability to create the contents in which to facilitate their own learning.[11] Additionally, d) Learned skills and functions are not separate and isolated learned in a linear fashion, but rather they consist of networks of interrelated functions in which complexity rather than difficulty should be emphasized.[12] Also learning both concepts and behaviors is predicated on the coordination and building of basic skills that will parlay into increasingly more complex learned behaviors [13] and, e) regression is a necessary feature of learning.[14]

The internet has also provided us with the ability, using the knowledge-based of cognitive neurosciences, to create new models of human cognitive development that ought to amend our understanding of the learning process and how training is provided. There exists a relationship between context and performance [15] and from reflexes to abstract thinking with basic skills that integrate within more complex skills.[16] As a result, we often see skills regression with significant implications for how school programming and curriculum are implemented and assessed. The current but old linear models, consisting of the training of isolated skills, treat and measure learning success as a steady rise. It is even the basic assumption of IQ testing. [17] We will see how we likely will fare better by focusing less on answers to questions and more on why and how learners learn in the way that they do. Grading systems reflect the current thinking in education that certainly does not account for the necessary regression of skill. If the assessment is continuous, for example, rather than provided at some endpoint, a regression could then be accounted for with skills assessment then being individualized, functioning then in a process-oriented rather than linear fashion.

In providing some direct examples of how cognitive neurosciences can readily be applied to the classroom we can note that cognitive tutoring may be effective in providing innovative and practical software for everyday use.[18] Separately, we know that sleep patterns change as a function of development [19] Adolescents tend to sleep longer than do other age groups consequently affecting early morning cognitive capacity. Additionally, the circadian rhythms of teenagers are different when compared to other age groups. School and class scheduling needs to account for that. Although repetition of material is necessary for effective learning [20] the cognitive neuroscience literature has noted the importance of a “spacing effect” where we have learned that students retain significantly more material when learning sessions are spread over time as opposed to being provided in a single session.[20] The consequence is to provide variety in the classroom with material reiterated over the course of a semester rather than in a session or in a few days. We know that each of us possesses similar brain anatomical structures, but the neural ensembles associated with that anatomy function in a highly individualized manner. [3,12] Therefore, learning tools that are adjustable to the needs of an individual learner are particularly valuable in the classroom. We will later see that if “you don’t use it you lose it”. However, we know that individuals who read more challenging books possess a significantly greater variety and number of neural connections. This research can be practically applied to the classroom in obvious ways.[21]

Other examples of translational neuroscience for classroom applications would include the relevance of making learning a positive experience which is in turn related to the release of dopamine, which in turn aids in memory for facts.[22] The relation between movement and cognition associated with the release of brain-derived neurotrophic factor (BDNF), associated with neuroplasticity and associational networks, discussed in greater detail later, is yet another example of how the marriage of cognitive neurosciences and education may be applied.[23]

Education researchers are beginning to approach complicated functions such as memory and learning physiologically. [3,4] We think that the knowledge-base of the neurosciences, Human Factors, and even Industrial Engineering that address notions of efficiency, can offer benefits to education throughout a learner’s life. [5,6] Recent research on the human brain is providing us with a better understanding of the processes of human learning in the classroom and therefore improving teaching methods. [4,7,8,9]"

Comment 6:

What does this mean?  (Line 117)(Education has been grabbing at straws for a long time)

Removed

Comment 7:

Nice example. Is there also one from the school years? (reference the State of Georgia mandating Mozart  listening)

I have added a paragraph (lines 137-142) on the ineffectiveness of non-evidenced-based educational mandates.

"Even worse than the example cited above are government-ordered educational mandates. In 2010, for example, the US State of California had greater than 50 of such mandates that required specific activities to be performed by school districts at a collective cost then of over 400 million dollars. Many of these requirements did not benefit students or educators. The existing mandate system also can reward districts for performing activities not only inefficiently but ineffectively"

Comment 8:

Though I don't doubt it, do you have evidence to cite here?("Most currently prevailing approaches to teaching poses a significant bias to the function- ing of the left hemisphere.")

citation in text lines (144-145)

Comment 9:

Inference rather than 'clever guessing'?

changed in text (line 148)

Comment 10:

Evidence that this is not true in 'eastern' cultures?

cited

Stigler JW, Lee SY, Stevenson HW. Mathematics classrooms in Japan, Taiwan, and the United States. Child development. 1987 Oct 1:1272-85.

Also

Spiegel A. Struggle for smarts> How eastern and western cultures tackle learning. 12 November 2012 NationalPublic Radio (https://www.npr.org/sections/health-shots/2012/11/12/164793058/struggle-for-smarts-how-eastern-and-western-cultures-tackle-learning) (accessed 11 September 2022)

Comment 11:

Is this all a quote from Sperry?

Whoops! Yes it is and it is now in quotes (lines 158-165)

Comment 12:

Can you draw out an example from those cited of active inhibition of creativity?

I have added the following to the text to provide thinking on the subject of active impression. (see lines 167-180)

Educational institutions have placed a great value on verbal/numerical skills and categorization and have methodically disregarded those proficiencies that would support young children's development of imagination, visualization, and/or sensory/perceptual abilities. [23, 24,25] Dylan Williams [RR] indicated that in his review of the literature, practiced and trained skills do not typically generalize as being significantly related to specific training. He stated, “It is certainly unhelpful, and probably wrong, therefore, to talk about ‘critical thinking skills’. Critical thinking is an important part of most disciplines, and if you ask disciplinary experts to describe what they mean by critical thinking, you may well find considerable similarities in the responses of mathematicians and historians. The temptation is then to think that they are describing the same thing, but they are not. The same is true for creativity. Creativity is not a single thing, but in fact a whole collection of similar, but different, processes. Creativity in mathematics is not the same as creativity in visual art. If a student decides to be creative in mathematics by deciding that 2 + 2 = 3, that is not being creative, it is just silly since the student is no longer doing mathematics…Creativity involves being at the edge of a field but still being within it.

Comment 13:

 I wonder if you need to explore this idea of a strict left/right dichotomy.

I would rather not inasmuch as I do not quite think that there is a rigorous right/left dichotomy but rather association networks that are formed by effective right/left hemisphere communication (e.g. network differences in the embodiment of language). I don’t want to get into fights but simply introduce the topic of neuroeducation.

Comment 14:

I'm not sure what the cultivation of transcendence would look like. (line 202)

Ok. Got the point.

I have removed transcendence. I guess I was hoping for something spiritual like, “Wow, only four substances that define life adenine, guanine, cytosine and thiamine”

Comment 15:

This seems like it might be a bit of a straw man argument. Has it been proposed that educational progress should be measured by examining regional cerebral differences? ("(A) the examination and study of regional cerebral differences in brain function as a way of explaining and evaluating the learning process within the educational system is a non-starter.")

The issue that will be developed further is based on a number of points, one of which s the example proved later in the text of a university student (patient) with congenital absence of most of the cortical gray matter whose networking based on regional cerebral blood flow was effective at problem-solving and verbal tasks at the highest level. Additional support for the notion of embodied networks rather than simple right/left differences and efficiencies of interregional communication are provided later. So I am not so sure that we are dealing with a 'straw man' argument here. Perhaps for the adult learner more so but apparently not for the younger learner (i.e. pre 23-26 years). I certainly do understand how reviewer 1 got to that point though.

Comment 16:

This is an interesting idea. ("(C) educational systems would be better to examine student performance and teach towards “cognitive efficiency” rather than simply mastery v. non-mastery. Educa- tors, although perhaps not palatable to conceive of early childhood education as such, are producing a product and production management techniques should be useful for evalu- ating not just the product but the process or “manufacture” of that product as well.)

Thanks

Comment 17:

Might be useful to bring in the idea of 'interactive activation' here (Mark Johnson)

Good point and done! (The addition between lines 229-238)

"Developmental cognitive neuroscience is a newly developing field that as a consequence of its newness has generated numerous hypotheses that can appear fragmentary in nature. For us to apply sometimes disparate hypotheses into practice could be confusing at best and potentially damaging at worst. We, however endeavoring, not to throw the baby out with the bathwater. 

With advances in aiming sciences, genetic, and electrophysiological approaches currently available we have begun to better comprehend surprising new findings applied to new situations including the classroom.[BB] Its is suggested that the reader refer to the theoretical review and hypotheses drawn by Mark Johnson who has covered the principal arguments in this burgeoning field copiously [BB]

These problems should not be a surprise. Developmental cognitive neuroscience is a newly developing field that as a consequence of its newness has generated numerous hypotheses that can appear fragmentary in nature. For us to apply sometimes disparate hypotheses into practice could be confusing at best and potentially damaging at worst. We, however endeavoring, not to through the baby out with the bathwater. With advances in aiming sciences, genetic, and electrophysiological approaches currently available we have begun to better comprehend surprising new findings applied to new situations including the classroom.[BB] Its is suggested that the reader refer o the theoretical review and hypotheses drawn by Mark Johnson who has covered the principal arguments in this burgeoning field copiously [BB]

Comment 18:

 It's not clear what optimization means here.

The term was applied earlier in the paper however, Optimization is an industrial engineering concept that relates to production management. It could best be understood in the context of efficiency utilizing the tools of linear programming. We had earlier referenced thinking of education as a product and that then would necessitate product management tools applied to the educational process. 

Comment 19:

It's not clear what this figure has been included to show.

The figure illustrates the fact that in early life the typical organizing patterns of brain areas are highly plastic. Meaning that the traditional understanding of the control functions of Brodmann’s areas is less relevant. It is known for example that in Rasmussen’s syndrome (a form of lateralized intractable seizures prior to age three, hemispherectomy may be performed on the child with little or no cognitive consequence and minor motor problems if any. So the point is that we are heading to build associational networks rather than examine the function of left v. Right or temporal-frontal connections or front-limbic connections but rather how to most efficiently bulled associational networks.

Comment 20:

I understand that you're demonstrating the importance of networks here, but it seems convoluted and there are no citations. 

Whoops again. Citation is easy. Me.
Added

Comment 21:

There's almost no evidence for this with respect to complex behaviour in humans. (According to Dehortrer and Del Pino [62] it is during a critical period that a relatively enduring alteration occurs that is required for an effectively operating brain function to occur. If that experience does not occur during a critical period it will likely not occur at mastery level in the lifetime of the individual. For example, learning ballet for the first time at age twenty or the violin at the same age.)

Well, yes and no. I take the point. and have thusly changed the term form critical to optimal periods. I have additionally amended the text (lines 374-384)

Reference music training it can be said that an optimal period and possiblly critical periods were seen in a study of violin training in which in a sample of 60 musicians and nonmusicians (i.e. those who started training before the age of 7 years exhibited increased corpus callosum size (Schlaug et al., 1995; Flohr and Hodges, 2006 etc).

The text now reads, "

According to Dehortrer and Del Pino [62] and others [DD, EE, FF] it is during an optimal period that a relatively enduring alteration occurs that is required for effectively operating brain function to occur. If that experience occurs during an optimal or sensitive period permanent differences can be found distinguishing, for example, musicians from non-musicians and from those trained early in life rather than later.[DD,EE] As sensitive periods of experience-dependent plasticity may allow for compensatory or alternative patterns of connectivity, it allows for the “tuning” of the execution of a given function by providing flexibility in functional network development. The specific school-based skills associated with critical and sensitive or optimal periods are represented in Figure 3."

Comment 22:

Based on? (figure legend 3. Figure 3. Sensitive periods during early brain development for (A) Sensation, emotional control, numerosity and symbolic representation, working and long-term memory and vocabulary.)

references added

Comment 23:

This paragraph needs citations. 

"Obviously there can be no learning without memory. What is learned requires encoding, integrated into memory and stored for further use. Learning and memory are necessarily linked in the brain. One has only learned something when it is coded in some form of memory for recall or use as a skill in the future. Learning then necessitates acquisition of information and the strengthening that information with previously acquired knowledge and its associated networks in memory. Memory is not a unitary proves but rather a collective of different process with different characteristics and processes. For example, there exists memory for skill, facts, and life events distributed and retrievable differently. Knowledge is represented in the brain through interconnectivities of networks. Recall and its associated retrieval processes are associated with spreading activation which then can lead to the recall of associated memories. Our brains then, in Hebbian terms, learn by making and strengthening connections through association and context. Knowledge is thought to be coded in the brain in an interconnected network, with similar or associated items and concepts being more strongly connected. Recall or retrieval of some fact, memory, or piece of knowledge will spread activation to help recall of other similar and associated memories or knowledge. These connections between neurons through the synapses are continually changing with learning and experience. Learning in the brain is the process of building and strengthening synaptic and regional connections and connectivities.

Now corrected including references and typos fixed:

Obviously, there can be no learning without memory. What is learned requires encoding, integrated into memory, and stored for further use. Learning and memory are necessarily linked in the brain. One has only learned something when it is coded in some form of memory for recall or use as a skill in the future. Learning then necessitates the acquisition of information and the strengthening of that information with previously acquired knowledge and its associated networks in memory. Memory is not a unitary process but rather a collective of different processes each with different characteristics. For example, there exists memory for skill, facts, and life events distributed and retrievable differently.[47, 50] Knowledge is represented in the brain through interconnectivities of networks. Recall and its associated retrieval processes are associated with spreading activation which then can lead to the recall of associated memories.[JJ] Our brains then, in Hebbian terms, learn by making and strengthening connections through association and context. Knowledge is thought to be coded in the brain in an interconnected network, with similar or associated items and concepts being more strongly connected.[JJ] Recall or retrieval of some fact, memory, or piece of knowledge will spread activation to help recall other similar and associated memories or knowledge. These connections between neurons through the synapses are continually changing with learning and experience. Learning in the brain is the process of building and strengthening synaptic and regional connections and connectivities.[KK, LL]

Comment 24:

This idea is complex and nuanced. If this is going to be explored it needs much more detail. (i.e "This is consistent with the notion that spe- cific concepts, are coded by distinctive activity patterns of in different sets of neurons.")

Of course, it is nuanced. Given the task and the point raised being very well taken, I have referred the reader to a larger review on the topic and indicated that the notion is, in fact, nuanced and requires a thorough analysis of the available literature. (lines 449-452)

Comment 25:

There's a lot of ideas in this section but it's not clear where it's heading and again there are no citations.

Citations provided and ideas reorganized.

Comment 26:

According to what pedagogical theory? ("This concept is most readily seen when examining the nature of embodied language., It is less important to understand that Broca’s area is in the frontal lobes, anterior to the Sylvian fissure is responsible for expressive language, and that the temporal lobes, and Wernicke’s area, in particular, in the left temporal region is responsible for receptive language. More to the point, as evidenced in Figure 4, is that the networks are different for leg, face, or arm rerated words.[UU] Education’s function is to build associative networks of the kind evidenced in embodied language.[VV,WW])

references added to pedagogical issues. (lines 491-497)

Solé RV, Corominas‐Murtra B, Valverde S, Steels L. Language networks: Their structure, function, and evolution. Complexity. 2010 Jul;15(6):20-6.

Willis J. Review of research: Brain-based teaching strategies for improving students' memory, learning, and test-taking success. Childhood education. 2007 Aug 1;83(5):310-5.

Gallese V. A neuroscientific grasp of concepts: From control to representation. Philosophical Transactions of the Royal Society of London. Series B: Biological Sciences. 2003 Jul 29;358(1435):1231-40.

Comment 27:

This is not a metric I've come across, where is your evidence for this?

(lines 498-520) The text has been amended.

Great statement. It is an example actually of why developmental cognitive neuroscience needs to, among other things, address classroom learning and developmental issues. Mathematics as well as industrial engineering have been involved in this issue for hundreds if not thousands of years. The metric and its application have not been particularly applied in the behavioral sciences and education with perhaps the exception of Human Factors Psychology.

In mathematics, people study the lengths of paths to construct short paths. It is often useful to find the shortest path. The shortest path is a path between two nodes that has the fewest edges if the cost (a measure of how much effort it takes to travel along an edge in a network. In real life, cost may measure distance, time, or something else) of traveling along each edge is the same (for example, if each edge is a street of the same length). More generally, given an origin node and a destination node, the shortest path (a path from the original node to a destination node that has the lowest total cost among all the paths from the origin to the destination) from the origin node to the destination node is a path that has the lowest total cost among all paths from the origin to the destination. So the young lady with the congenital hydrocephalus in Fig 1. Is. More efficient processor than a young school-aged child and the adult brain has more efficient processing than that of a school-aged child as there is a relative paucity of connections in the adult compared with the young child. In other words, pruning, by this standard, makes the brain more efficient (or optimized in production management terms).’

(Newman, M. E. J. 2018. Networks, 2nd Edn. Oxford, UK: Oxford University Press..)

Comment 28:

(Reference Figure 4 )Are these data or just representative of an idea?

Real data

Comments 29 and 30:

29. Are the facts presented in educational settings 'unrelated'?

30. It seems that learning facts can be a part of associative learning, just like any other sort of information. ("One can conclude up to this point in the discussion that rote learning and memorization of unrelated facts is not efficient as it is, generally speaking, not the way the brain forms associative networks for effective learning. Actually, the brain learns by forming and strengthening connections between associated concepts so that learning in context and linking and forming associations with allied knowledge will be more effective for retention. This is seen very clearly in the benefit of using mnemonics that can greatly help learning by creating meaningful associations between elements. Using the phrase “’i’ before ‘e’ except after ‘c’” is an example of a mnemonic to remember the correct order of the ‘i’ and the ‘e’ in the word ‘friend’.)

Not so well without a) meaning and b) self-reference c) connections to other factoids d) language in non-procedural learning e) novelty f) attentional focus g) time etc.

Comment 31:

This is interesting but feels like a bit of a tangent. (Physical Activity and Cognitive Relationships).

Understood. What seems to be missing is the direct relevance to human rather than animal models and that I will fill in.

The purpose of the paper is to overview the other topical areas under the heading of, "The “Brain Goes o School” and to outline areas where the burgeoning field of developmental cognitive neuroscience can play a role.

Comment 32:

Figure 5 Are these figures re-produced with permission?

Yes and now so indicated.

Comment 33 and 34.

Citations? (lines 577-606)

Done!

Comment 35:

It's difficult to synthesise all of this!

This paragraph is a list serving as a summary of what is known in developmental cognitive neuroscience that may be sitting undid learning. It is now referenced and hopefully, the reader will available themselves of the collection of papers in the special issue and hopefully of the now appropriately referenced literature. 

Comment 36:

Citations? (section 8.2)

Now provided

Comment 37:

This is not always true. Some types of learning can occur without attention. ("Before individuals can learn or have a skill rehabilitated, something or someone, must cap- ture their attention.")

Agreed especially as it relates to subliminal stimulation (e.g. posters hanging on a wall in a driving school and then subjects asked to recall information that was only represented by the posters etc). The sentence has been modified accordingly.

Comment 38:

Why is this? ("Emotional responsivity, thinking and learning need to be linked.")

References added to text and…

(Cf. Panksepp 1998) "Emotions are the psychoneural processes that are influential in controlling the vigor and patterning of actions in the dynamic flow of intense behavioral interchanges between animals as well as with certain objects that are important for survival. Hence, each emotion has a characteristic “feeling tone” that is especially important in encoding the intrinsic values of these interactions, depending on their likelihood of either promoting or hindering survival (both in the immediate “personal” and long-term “reproductive” sense). Subjective experiential-feelings arise from the interactions of various emotional systems with the fundamental brain substrates of “the self,” that is important in encoding new information as well as retrieving information on subsequent events and allowing individuals efficiently to generalize new events and make decisions.
Panksepp went further to propose seven primary emotional systems/prototype emotional states, namely SEEKING, RAGE, FEAR, LUST, CARE, PANIC/GRIEF, and PLAY that represent basic foundations for living and learning.

The mechanisms are discussed more fully in Tyng CM, Amin HU, Saad MNM and Malik AS (2017) The Influences of Emotion on Learning and Memory. Front. Psychol. 8:1454. doi: 10.3389/fpsyg.2017.01454.

Comment 39:

'Neureucators' What does the term mean?

School teachers with a grounding in developmental cognitive neuroscience.

Text amended to insure greater clarity.

Comment 40:

There's plenty of procedural learning in school. Indeed the first few years are almost exclusively procedural. (referenced to the text "In classrooms, educators largely teach to semantic memory. [122, 123, 124] It is a less optimized form of instruction since there exists less control over the input in the nervous system.)

I have added the qualifier, “when sitting in class”.

Reviewer 2 Report

The article is very well organized and runs through several issues of great importance to educational neuroscience. I have pointed out some typographical errors in the text.

One issue that could be modified is to include research from human studies instead of animals. Especially the second example (page.10-11) could be determined with data from human research.

The second issue that needs attention is the reference to learning styles (p. 15). There is rich literature on the subject, and the reference in this particular article is inconsistent with the findings of educational neuroscience. Here the need to debunk neuromyths could be argued.

Finally, it is suggested that the table on page 15 be diagrammatically made as a process.

Author Response

The article is very well organized and runs through several issues of great importance to educational neuroscience. I have pointed out some typographical errors in the text.

Thank you for doing so.

One issue that could be modified is to include research from human studies instead of animals. Especially the second example (page.10-11) could be determined with data from human research.

The paper has been significantly revised and human experiments relevant to the arguments presented have been included in the revised manuscript and marginalized corrections are attached.

The second issue that needs attention is the reference to learning styles (p. 15). There is rich literature on the subject, and the reference in this particular article is inconsistent with the findings of educational neuroscience. Here the need to debunk neuromyths could be argued.

I completely agree with the comment. Given the fact that this paper endeavors to overview the other papers in this special issue, and the neuromyth of learning style did not come up, I took the reviewer's suggestion and changed the term from learning "style" to "patterns".  There the reference is more on the development of associational networks and that is now heavily justified in the paper.

Finally, it is suggested that the table on page 15 be diagrammatically made as a process.

Thanks for the suggestion and this has been done.
